# Precision Health Care Elements, Definitions, and Strategies for Patients with Diabetes: A Literature Review

**DOI:** 10.3390/ijerph18126535

**Published:** 2021-06-17

**Authors:** Satriya Pranata, Shu-Fang Vivienne Wu, Javad Alizargar, Ju-Han Liu, Shu-Yuan Liang, Yu-Ying Lu

**Affiliations:** 1School of Nursing, National Taipei University of Nursing and Health Sciences, Taipei City 112, Taiwan; satriya.pranata@unimus.ac.id (S.P.); juhan@ntunhs.edu.tw (J.-H.L.); shuyuan@ntunhs.edu.tw (S.-Y.L.); yuyin@ntunhs.edu.tw (Y.-Y.L.); 2Faculty of Nursing and Health Sciences, Universitas Muhammadiyah Semarang, Kota Semarang 50273, Central Java, Indonesia; 3Research Center for Healthcare Industry Innovation, National Taipei University of Nursing and Health Sciences, Taipei City 112, Taiwan; jaz.tmu@gmail.com

**Keywords:** personal health services, diabetes mellitus, glycemic control, patient care team

## Abstract

Diabetes is a prevalent disease with a high risk of complications. The number of people with diabetes worldwide was reported to increase every year. However, new integrated individualized health care related to diabetes is insufficiently developed. Purpose: The objective of this study was to conduct a literature review and discover precision health care elements, definitions, and strategies. Methods: This study involved a 2-stage process. The first stage comprised a systematic literature search, evidence evaluation, and article extraction. The second stage involved discovering precision health care elements and defining and developing strategies for the management of patients with diabetes. Results: Of 1337 articles, we selected 35 relevant articles for identifying elements and definitions of precision health care for diabetes, including personalized genetic or lifestyle factors, biodata- or evidence-based practice, glycemic target, patient preferences, glycemic control, interdisciplinary collaboration practice, self-management, and patient priority direct care. Moreover, strategies were developed to apply precision health care for diabetes treatment based on eight elements. Conclusions: We discovered precision health care elements and defined and developed strategies of precision health care for patients with diabetes. precision health care is based on team foundation, personalized glycemic target, and control as well as patient preferences and priority, thus providing references for future research and clinical practice.

## 1. Introduction

Diabetes is a chronic disease and a major health concern in modern society. diabetes is typically characterized by an abnormal increase in glucose levels caused by one or two mechanisms: inadequate insulin production by the pancreas or inadequate cell sensitivity to the action of insulin caused by the reduced function of insulin receptors. In 2019, an estimated 463 million patients had diabetes. By 2045, this number is predicted to increase to 700.2 million [1].

Other data indicate that people with diabetes have a greater risk of experiencing various complications than other individuals. People with diabetes are 2–3 times more likely to develop cardiovascular diseases and up to 10 times more likely to develop end-stage renal disease and part or complete amputation of a lower limb; further, an amputation is performed somewhere in the world every 30 s. In 2019, the total health care expenditures on diabetes for people aged 20–79 years were an estimated US$760 billion, such expenditure may reach US$825 billion in 2030 and US$845 in 2045 [1,2].

Hospital- and community-based interventions to avoid diabetes-related complications have been based on evidence-based practice and guidelines for diabetes care. A guideline typically provides a set of recommendations along with eligibility criteria that restrict their applicability to a specific group of patients for disseminating such knowledge and standardizing care to ensure the highest quality of care [3]. Furthermore, interventions based on hospital guidelines involve the promotion of self-efficacy, health education, self-management, and health coaching [4,5]. Guidelines and evidence-based practice are crucial for defining the quality of care. Nevertheless, in certain situations, deviating from such guidelines and practice is desirable and helps address the needs and peculiarities of patients with diabetes. Specifically, with the availability of health data related to patients with diabetes, precise identification based on treatment demands and targets can be executed, posing challenges on diabetes care to clinical guideline recommendations. However, studies related to preference regarding integrated individualized health care in diabetes are limited.

Research efforts in this direction are termed “precision medicine”. The vision of precision medicine is that this medication is predictive, preventive, personalized, and participatory [6,7]. Moreover, through precision medicine, doctors and researchers adopt treatment and prevention strategies more accurately and consider differences between individuals rather than the one-size-fits-all approach [8]. As an individual’s experience of both health and disease is unique at the molecular, cellular, and organ levels, treating the causes rather than the symptoms of diseases is achievable [9]. The precision medicine approach is becoming a trend in clinical settings, especially after former US President Obama’s launch of the Precision Medicine Initiative in early 2015. The primary aim of precision medicine is to improve clinical outcomes for individual patients through precise treatment targeting by leveraging genetic, biomarker, phenotypic, or psychosocial characteristics that distinguish a given patient from others with similar clinical presentations [3].

In addition, the terminologies defined above can be applied to other approaches in the form of precision health care (PHC). PHC involves patient care preferences, patient-oriented care, evidence-based care, and self-management [10,11,12]. PHC is a care delivery model that relies heavily on data, analytics, and personal information [11,13].

Research has increasingly explored the diagnosis, treatment, and disease prevention benefits of personalization, precision medicine, and PHC, but little information regarding PHC in the diabetes population is available. The objective of this study was to discover PHC elements, derive definitions, and develop strategies for treating a person with diabetes by using a two-stage innovative process involving a systematic review of scientific methods. The first stage involved a literature search with a careful evaluation of evidence and extraction and integration data. The second stage entailed discovering PHC elements, deriving definitions, and developing strategies aimed at providing a more comprehensive approach to health intervention and disease management in clinical settings. We explored additional health care literature and international trends and integrated them with our literature review to promote person participation, foster good health behavior, reduce complications, and reduce the family care burden and medical costs. The discovery of PHC elements and definitions and strategy development for people with diabetes are expected to change traditional clinical intervention, decision-making, and future research trends.

## 2. Materials and Methods

We conducted a search on an international database to identify articles published from January 2014 to March 2020. We included studies that (1) were published in a qualitative or quantitative research journal with peer review; (2) were published between January 2014 and March 2020; (3) involved a literature review, meta-analysis, guideline, cohort, or randomized controlled trial (RCT); (4) were published in English for full text; (5) were related to precision medicine and PHC in a diabetes population; and (6) provided conclusions that clearly suggested specific application for patients with diabetes. We excluded studies that (1) included nonhuman groups as research objects; (2) did not study PHC and precision medicine for diabetes; (3) were duplicate entries in the search results; (4) were incomplete; (5) had yet to be disclosed as published, or (6) were not published in English.

The literature review involved a document analysis that was implemented in two stages.

### 2.1. First Stage

This stage comprised a literature search through a primary search strategy, a secondary search through a study quality review, and a study extraction.

#### 2.1.1. Primary Search Strategy

Based on population, intervention, comparison, and outcome, we used the Boolean logic operators “AND” and “OR” and different intersections to combine the following keywords: precision medicine and diabetes, precise health care, precise care, precise health, personalized care, integrated care, patient-centered, patient-oriented care, precision healthcare, precision nursing, individualized care, and patient priority-directed care.

We identified articles in the literature that clearly focused on precision medicine or PHC for patients with diabetes, with a clear reference in the title and abstract. Moreover, we focused on articles published in English because their accessibility to the broad scientific community enhances reproducibility. This search yielded 1337 relevant articles that included the words precision medicine, PHC, and diabetes in the title or abstract from January 2014 and March 2020. Members of the study team reviewed the title and abstract, further excluded the articles based on the following criteria: duplication (*n* = 69), non- English publication (*n* = 38), incomplete articles (*n* = 96), participants in the article not having diabetes (*n* = 188), and non-inclusion of PHC and precision medicine concepts (*n* = 886). Accordingly, the number of articles dropped were (*n* = 1277) after the application of the aforementioned exclusion criteria. We obtained the full texts of all 60 articles that met the inclusion criteria (Figure 1).

#### 2.1.2. Secondary Search Strategy by Using Study Quality Review

Two experts reviewed each article by using the Joanna Briggs Institute Validity Scale 2011. We reviewed the quality of all 60 articles. The Critical Appraisal Skills Program was used to evaluate article quality. Finally, 35 articles were included in the analysis (Figure 1). Articles were selected through a consensus review and then extracted and integrated.

##### Study Extraction of Elements on PHC for Patients with Diabetes

The literature search results were compiled and are presented in Table 1.

Each article in Table 1 was read by five members of the research team. Every section that mentioned PHC for patients with diabetes was marked. We divided any essential part and suggestion in each article in Table 1 were generally delineated by punctuation. Furthermore, based on the study protocol, we developed coding roles and coding sheets that included a list of mutually exclusive and exhaustive concepts (e.g., interdisciplinary collaborative practice and personalized genetic or lifestyle factors). Some of these coding categories represent PHC elements for patients with diabetes. Five experts separately coded, extracted the data, and integrated the results; furthermore, they reached a final consensus after discussion.

### 2.2. Second Stage

This stage entailed discovering PHC elements, deriving definitions, and developing clinical strategies for patients with diabetes.

First, we summarized the derived literature search results in Table 1 to analyze the PHC elements for diabetes. Second, we discussed the concept description of each element, integrated the data in Table 1 into the operational definition, and then developed strategies for clinical application of the elements.

## 3. Results

This section may be divided into subheadings. It should provide a concise and precise description of the experimental results, their interpretation, as well as the experimental conclusions that can be drawn.

### 3.1. Analysis and Compilation of the Derived Literature Search

After comprehensively analyzing 35 relevant empirical studies, we summarized the PHC elements for patients with diabetes. The most frequently used PHC elements for patients with diabetes are as follows: personalized genetic or lifestyle, biodata- or evidence-based, glycemic target, patient preferences, glycemic control, interdisciplinary collaboration practice, self-management, and patient priority direct care. Therefore, we first clarify the definitions and operation of the PHC concept for patients with diabetes. Through this process, the structure of the concept can be clarified.

### 3.2. Discovery of the Elements and Their Concept Description

We discovered the PHC elements for diabetes through the literature study and discussed the concept description of eight elements. We identified the concepts by extracting the data in each article, comparable definitions used in precision medicine and health then integrated the results. Therefore, we reached a final consensus after discussion with the research team. The concept description and strategy of each element can be seen in Table 2 and are explained below.

#### 3.2.1. Personalized Genetic or Lifestyle

With the advances in medical technology, various new methods continue to emerge, including the trend of genetic testing or genomic test to determine a person’s genetic profile. Genetic or lifestyle analysis helps delineate complication risk, glycemia tolerance, side effect of diabetes treatment through gender analysis, age, comorbid condition, cholesterol, and blood pressure [16,32]. The genomic test can be used for screening for diabetes autoantibodies remains drug or insulin dose. Furthermore, a genomic test is performed by observing the gene encoding glucokinase, *HNF1A*, and *HNF4A*, which are associated with diabetes onset [25]. The patient genotype influences response to the side effect of diabetes treatment, and C-peptide is a biomarker to guide treatment choice (insulin deficiency). Moreover, single-nucleotide polymorphism provides information regarding drug toxicity [15,16,25]. Genetic or lifestyle factors can be assessed during PHC for diabetes by using risk prediction charts [46]. Furthermore, data on genotype or electronic health records from hospital records can be used to determine the most appropriate diabetes care for patients [11,47].

#### 3.2.2. Biodata- or Evidence-Based Requirement

Health profiling should further include other social and environmental data of the patient. The PHC approach can provide precise therapy to patients based on their genetic characteristics [8]. Moreover, the genetic examination can detect various potential health problems, including cardiovascular disease and a person’s metabolic ability toward a nutrient. Furthermore, electronic health medical records and health insurance databases contain rich clinical information, including data from remote sources to identify disease risk factors through computation and bioinformatic methods [27,36].

#### 3.2.3. Glycemic Target

According to American Diabetes Association (ADA) guidelines, target and therapy differ based on the features and responses of each individual namely, (A1c, Blood pressure, and Cholesterol; Paschou & Leslie, 2013; Subramanian & Hirsch, 2014). Factors that must be considered in PHC are patient age, individual treatment goals, HbA1c on current clinical data, lifestyle, and physical activity of patients [18]. The shared decision-making assessment tool can be used to assess patient goals based on their preferences [17].

#### 3.2.4. Patient Preferences

The specific treatment approach must be individualized based on patient-specific factors and preference, [15]. Using the shared decision-making tool, patients’ need for additional medication and their concerns regarding hypoglycemia and hyperglycemia can be further evaluated [15,17].

#### 3.2.5. Glycemic Control

A more potent drug can be used to achieve a greater reduction in HbA1c to <6.5% [18]. Individual GTs based on ADA consensus in addition to patient-specific factors are approximately <6.5%, <7%, and <8% [24]. In Precision health care, participant and facilitator will be setting a goal and developing a detailed written action plan for achieving glycemic control in reducing HbA1C. All medication and nursing management of patients will be tailored based on patient needs, priorities, and preferences. Patient difficulties, frequency, and intensity of intervention for the action plan will discuss with patients.

#### 3.2.6. Interdisciplinary Collaboration Practice

Interdisciplinary collaboration can be accomplished if all professionals from different backgrounds learn to work together as a part of the health care team [21]. Teamwork entails discussion to decide on the most appropriate treatment for patients, which can then be shared through decision-making tools [16,26,33].

#### 3.2.7. Self-Management

Diabetes self-management education, self-efficacy enhancing intervention program, self-management are ongoing processes comprising strategies to help individuals with chronic conditions, and their families and caregivers, to better understand and manage their illness and enhance health behavior, [48]. Self-management in PHC includes focusing on self-identified needs or problems that require continual monitoring and taking appropriate actions [4]. Individualizing therapy allows patients to effectively self-manage their disease through increased self-efficacy [4,22].

#### 3.2.8. Patient Priority Direct Care

We can improve outpatient care decisions through improved support systems that prioritize care recommendations and enhance communication of treatment-relevant information to patients with diabetes [27]. For example, through shared decision-making in clinical practice, patient condition can be improved through consideration of the whole individual, including the complex interplay of comorbid conditions, psychosocial and functional status, and individual needs [22,27]. Table 2 presents the elements, concept description, and clinical strategies on PHC for diabetes.

### 3.3. Definition of PHC

We integrated the elements in the literature for an operational definition of PHC and PHC for patients with diabetes as below.

We identified six elements of PHC, namely purpose of health, care preferences, patient priority direct care, interdisciplinary collaborative practice, popular biodata- or evidence-based practice, and patient self-management [10,11,13,21]. Therefore, we defined PHC as providing integrated care based on individual needs, including interdisciplinary cooperation and patient involvement in decision-making regarding health goals, providing care that meets patient expectations and preferences, providing patient-oriented care, and using biodata as evidence-based care disposal to improve patient self-management.

Moreover, based on the comprehensive literature analysis and data integration, we summarized that personalized genetic or lifestyle, biodata- or evidence-based, glycemic target, patient preferences, glycemic control, interdisciplinary collaboration practice, self-management, and patient priority direct care were the core elements for patients with diabetes. Furthermore, the operational definition of PHC for diabetes was tailoring integrated care through interdisciplinary collaborative practice among patients, nurses, and physicians based on the patient’s genetics or lifestyle, glycemic target, biodata- or evidence-based practice, patient preferences, and priority for improving patient self-management to achieve glycemic control.

### 3.4. Strategies of PHC for Diabetes in Clinical Practice

After discovering the PHC elements and defining them for diabetes, we developed strategies for the application of these elements for treating diabetes in clinical practice. The strategies can be elaborated as follows:

SDM became crucial in PHC because, through this method, patients with diabetes have an opportunity to choose either insulin injection or oral drugs for glucose control; furthermore, health professionals must discuss the best suitable treatment with patients. The shared decision-making process, which combines knowledge, communication, and respect to establish a glycemic target and medical consensus, is based on patients’ preferences and values. Shared decision-making can be achieved by using the shared decision-making tool for assistance to implement PHC. When analyzing elements of PHC for diabetes, clinical strategies or intervention for clinical application is essential. The clinical strategies or interventions are illustrated in Figure 2.

Figure 2 shows an example and the process of applying PHC for diabetes based on eight elements. Figure 2 explains the process in a real condition at a hospital or a clinical setting, wherein patients often show HbA1c values ≥ 6.5% despite taking two types of medication namely oral drug medication and insulin injection. The patient and health professionals should discuss and analyze the most appropriate treatment using the following five steps, which forms the acronym SHARE: (1) Seek patients’ participation by describing the problem clearly and openly to make the patients understand that a decision needs to be made; (2) Help patients explore and compare treatment options; (3) Assess the patient’s values and preferences; (4) Reach a decision with the patient; (5) Evaluate the patient’s decision of using insulin injection, maintaining oral medication, or no treatment change [49]. In addition, patients must be encouraged to use a more potent drug to achieve a greater reduction in HbA1c (<6.5%) based on patient-specific factors (age, individual treatment goals, HbA1c ADA guidelines, current clinical data, diet, lifestyle, and physical activity). It cannot be overstated that PHC allows patients to effectively self-manage their disease by increasing self-efficacy. Result outcomes can be improved through the promotion of patient participation, good health behavior, and reduced complications, family care burden, and medical costs.

## 4. Discussion

The main result of this study was the discovery of PHC elements. To apply PHC elements for diabetes treatment, patients must explain the pressing problem to health professionals by using shared decision-making tools [13,49]. The discussion between patients and health professionals will likely lead to a presentation of options [13]. The discussion of options between health professionals and patients can ensure agreement on the benefits, risks, costs, including patients’ beliefs, values, lifestyles, and glycemic target [29,37]. The information gathered evidence patient preferences [50].

Self-management was also a critical component of PHC for diabetes. Self-management involves prioritizing a precise treatment for patients [47]. Self-efficacy drives patients through self-management to follow the plans (e.g., test, medication, procedure, behavior change) formulated in collaboration with health professionals [48]. For self-management, the understanding of facts or perspectives among patients with diabetes is needed [51]. Verifying and clarifying this understanding is paramount for health literacy. Obviously, the decision of applying self-management for solving the problems is not always made when the problems are first discussed. Thus, health professionals must arrange follow-ups to track the outcome of decisions that have been made together. Several RCTs have reported the application and effectiveness of self-management in the PHC approach. Davies and Krag demonstrated that self-management as one of the PHC elements is effective in patients with chronic diseases such as diabetes; the concept is appropriate for these patients because it prioritizes precise problem solving and reduces unnecessary interventions. Furthermore, gender considerations are becoming increasingly essential [12,23]. Therefore, empirical research is required to verify the effectiveness of this concept.

There is a fundamental difference in diabetes care of type 1 and type 2. For type 1, insulin injections are needed generally. If the appropriate gene defect is identified, insulin injections can be stopped and replaced with oral sulfonylurea drugs [43]. New technologies, genetics, and next-generation DNA sequencing methods have the greatest effect on identifying important prognostic and treatment implications for type 1 diabetes [8]. Furthermore, genomics, environment, and lifestyle, are factors that influence the health of people with type 2 diabetes [30]. However, lifestyle factors play a big effect on prognosis [47].

In addition, patients with diabetes have a high risk of mortality due to COVID-19. Study of biological characteristics will determine the magnitude of the phenotypes and prognostic markers. Therefore, further study on which sub subgroups of patients with diabetes are expected to benefit the most from specific antiviral, immunomodulatory, and other treatment strategies in the context of patient precision medicine, which emerges as an urgent priority in the era of COVID-19 [52].

The other core elements of PHC for diabetes are based on professional team care, including medical specialists, nurses, nurse specialists, pharmacists, and nutritionists. In addition, patients are involved in decision-making and the preparation of care programs and goals. Every professional should implement interdisciplinary communication platforms and cooperation models [53]. A patient-centered care model can improve the current model of PHC, which can provide exclusive care for individuals with diabetes and enable establishing a treatment plan through improved accurate diagnosis and care, thereby improving treatment and care effectiveness. The PHC concept enables providing care for high-quality or customized-quality cases. Additionally, this care type considers not only medical care but also a patient glycemic target, glycemic control, biodata- or evidence-based patient preferences, and patient priority direct care [16,18,27,36,39]. Effective model design, innovation, and practical interventions ensure favorable patient outcomes and improved self-management [4,12,22,24].

Care programs and goals for an individualized glycemic target among patients on PHC are based on an agreement between health professionals and patients [24]. We can use recommended strategies, such as the ADA [18,24,34]. The individualization target was based on hypoglycemic episode risk among patients. We observed a concordance of ADA strategies with regards to patient classification; however, the concordance between these strategies is based on low hypoglycemic risk [24]. The type of hypoglycemic treatment based on ADA does not modify patient classification into different GTs. Using the conventional HbA1c target level of <7% (53 mmol/mol) [15,24], differences between patient glycemic targets can be considered based on whether the patient is managed through a restricted diet or pharmaceutical treatment (drug oral medication or insulin) [15,24,34,36].

Comprehensive research findings and recommendations are evident in current international care trends. Traditional general intervention in diabetes care is gradually being replaced by personalized PHC. However, some hospitals as service providers do not provide a PHC program, which includes staff preparation, a health system, and health workers, namely physicians, nurses, nutritionists, and other health professionals. Consequently, this aspect of PHC remains insufficiently developed.

Most studies were based on precision medicine for several chronic diseases such as diabetes and not PHC for diabetes. Although diabetes is a typical self-care or chronic disease, patients can live comfortably with the disease for long periods. PHC for diabetes involves the tailoring of care to fit individual needs. We hope that precision health strategies used for clinically applying the eight PHC elements for patients with diabetes will be humane and promote successful disease management.

We thus anticipate that additional research resources relevant to PHC for diabetes will be available in the future; such resources can enable researchers to robustly verify the PHC principle for patients with diabetes. The strategies of PHC for diabetes in clinical application must be patient-oriented, with dynamic personalized glycemic goals and care plans established through team care [4,5,21,22,24]. Based on the core PHC elements and the goals set by the patient and health team, intervention risks can be reduced, interdisciplinary integration can be developed among health teams, patient outcomes and quality of life can be improved, and unnecessary medical interventions can be avoided, thus reducing the medical burden. Therefore, we not only explored and established PHC elements for diabetes in a clinical setting but also developed strategies for applying PHC for diabetes. Furthermore, we suggested strategies for implementing PHC as follows: (1) assessment of complication risk by using risk prediction charts; (2) development of electronic health records in hospitals; (3) development of shared decision-making assessment tools; and (4) research on PHC with intervention focusing on the nursing practice. Integrating PHC with smart technology may facilitate the globalization of PHC programs for diabetes and other diseases. For successfully implementing PHC, more elements are needed for patients with diabetes and health professionals. Notably, strategies must be well researched before clinical application.

## 5. Conclusions

The PHC elements for diabetes are personalized genetic or lifestyle, biodata- or evidence-based practice, glycemic target, patient preferences, glycemic control, interdisciplinary collaboration practice, self-management, and patient priority direct care. Moreover, definitions and strategies were developed for applying PHC elements for diabetes based on eight elements which a team foundation, with personalized glycemic target and glycemic control as well as patient preferences and patient priority direct care as the basic principle. The elements and strategies can provide a reference for establishing PHC programs for diabetes in the future. We hope that future studies will evaluate the effect of PHC on diabetes and develop strategies for generalized usage.

## Figures and Tables

**Figure 1 ijerph-18-06535-f001:**
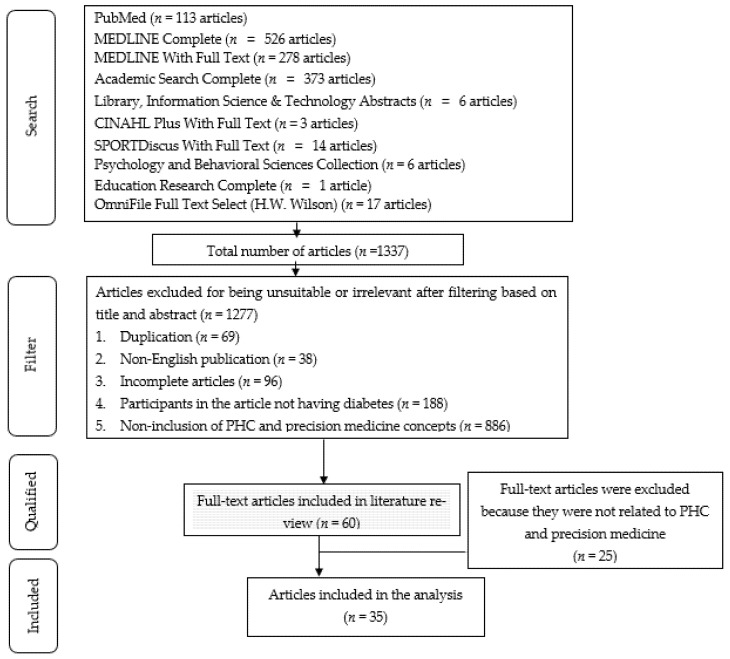
Flowchart of screening articles on PHC for diabetes.

**Figure 2 ijerph-18-06535-f002:**
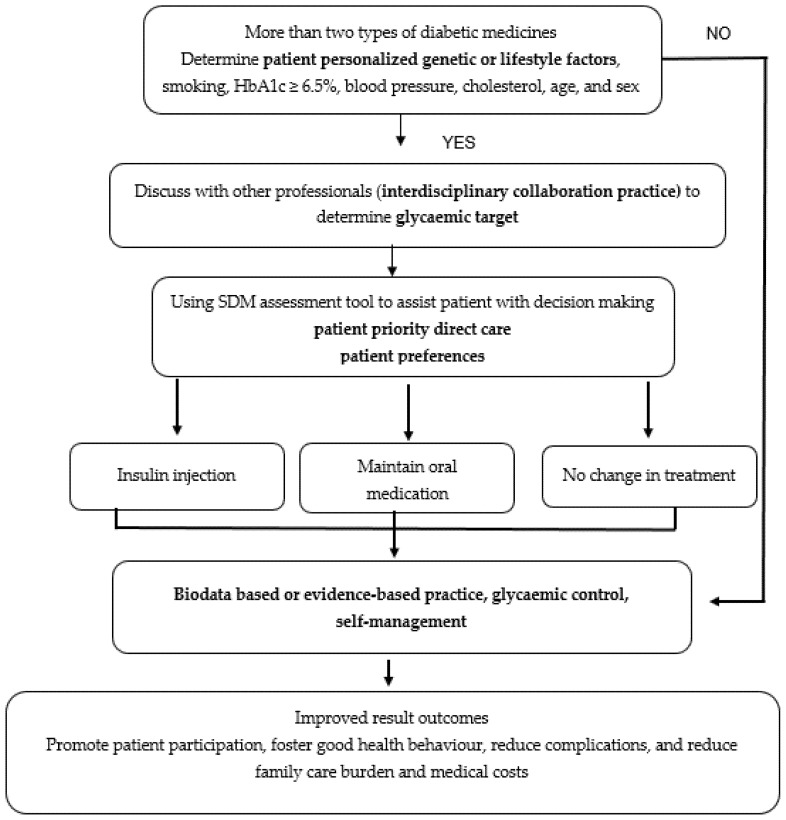
Strategies to clinically apply eight PHC elements for treating patients with diabetes.

**Table 1 ijerph-18-06535-t001:** Extraction of PHC Elements for Patients With diabetes.

No		Method	Population	PG/LS	BB/EB	GT	PP	GC	ICP	SM	PRDC
1.	Abbate, Mannucci, Cioni, Fatini, and Marcucci (2012) [14]	LR	Type 1 and type 2 diabetes	×							
2.	Meneghini & Reid (2012) [15]	LR	Type 2 diabetes	×			×			×	
3.	Spiegel and Hawkins (2012) [16]	LR	Type 2 diabetes	×					×		
4.	Paschou and Leslie (2013) [17]	LR	Type 2 diabetes	×		×	×				
5.	Subramanian and Hirsch (2014) [18]	LR	Type 2 diabetes	×		×		×			
6.	Davies et al. (2015) [19]	RCT	Type 2 diabetes	×							
7.	Groop, (2015) [20]	LR	Type 1 and type 2 diabetes	×							
8.	Jameson and Longo (2015) [8]	LR	Type 1 and type 2 diabetes	×	×						
9.	Sexton (2016) [21]	LR	Interdisciplinary teamwork care						×		
10.	Sherifali et al. (2016) [4]	LR	Type 2 diabetes							×	
11.	Sherifali (2016) [22]	LR	Type 2 diabetes			×					×
12.	Krag et al. (2016) [23]	RCT	Type 2 diabetes	×							
13.	Miñambres, Mediavilla, Sarroca, and Pérez (2016) [24]	CSA	Type 2 diabetes			×		×			
14.	Pearson (2016) [25]	LR	Type 2 diabetes	×		×					
15.	Fradkin, Hanlon, and Rodgers (2016) [26]	LR	Type 1 and type 2 diabetes	×					×		
16.	Holt (2016) [27]	LR	Type 2 diabetes		×		×				×
17.	Florez (2016) [28]	LR	Type 1 and type 2 diabetes	×							
18.	Meyer (2016) [29]	LR	Type 2 diabetes				×				
19.	Arnett and Claas (2016) [30]	LR	Type 1 and type 2 diabetes	×							
20.	Scheen (2016) [31]	LR	Type 2 diabetes	×							
21.	Floyd and Psaty (2016) [32]	LR	Type 2 diabetes	×							
22.	Rich and Cefalu (2016) [33]	LR	Type 2 diabetes	×					×		
23.	Krinsley, Preiser, and Hirsch (2017) [34]	CHT	Type 2 diabetes			×		×			
24.	Sherifali (2017) [5]	LR	Type 2 diabetes							×	
25.	Mahato, Srivastava, and Chandra (2017) [35]	LR	Type 1 and type 2 diabetes		×						
26.	Mayor (2017) [36]	LR	Type 2 diabetes	×	×	×	×	×			
27.	Mutie, Giordano, and Franks (2017) [37]	LR	Type 2 diabetes	×	×						
28.	Fitipaldi, McCarthy, Florez, and Franks (2018) [38]	LR	Type 2 diabetes	×	×						
29.	Horwitz, Charlson, and Singer (2018) [39]	LR	Type 2 diabetes		×						
30.	Greener (2018) [40]	LR	Type 1 and type 2 diabetes	×		×					
31.	Burke, Trinidad, and Schenck (2019) [41]	LR	Type 2 diabetes	×							
32.	Mannino, Andreozzi, and Sesti (2019) [42]	LR	Type 2 diabetes	×							
33.	Mohan and Radha (2019) [43]	LR	Type 1 and type 2 diabetes	×	×						
34.	Leggio, Tiberti, Armeni, Limongelli, and Mazza (2019) [44]	LR	Type 2 diabetes	×	×						
35.	Prasad and Groop (2019) [45]	LR	Type 2 diabetes	×	×						

Note: LR: literature review; CSA: cross-sectional analysis; RCT: randomized controlled trial; CHT: cohort study; PG/LS: personalized genetic or lifestyle; BB/EB: biodata- or evidence-based; GT: glycemic target; PP: patient preferences; GC: glycemic control; ICP: interdisciplinary collaboration practice; SM: self-management; PRDC: Patient priority direct care.

**Table 2 ijerph-18-06535-t002:** Elements, Concept Descriptions, and Clinical Strategies on PHC for Diabetes.

No	Elements	Concept Description	Clinical Strategies
1.	Personalized genetic or lifestyle	- Genetic or lifestyle analysis; genomic test screening for diabetes autoantibodies that remain after a drug or insulin dose, gene encoding glucokinase, presence of *HNF1A* and *HNF4A* that are associated with forms of diabetes onset; C-peptide is a biomarker that can be used as a guide to treatment choice (insulin deficiency); single-nucleotide polymorphisms provide information regarding drug toxicity	- Assessment of risk of complication by using risk prediction charts, genotype, or electronic health records
2.	Biodata-or evidence-based	- Genetic examination to detect various potential health problems, cardiovascular disease, a person’s metabolic ability to a nutrient, and HbA1c target	- Electronic health records and ADA guidelines
3.	Glycemic target	- Based on ADA guidelines, target and therapy differ based on the features and responses of each individual (including HbA1c, blood pressure, and cholesterol)	- Shared decision-making assessment tool
4.	Patient preferences	- Identification of whether the patient needs additional medication and their concern regarding hyper/hypoglycemia, further expressing their decision	- Shared decision-making assessment tool
5.	Glycemic control	- Supporting the use of a potent drug to achieve a reduction in HbA1c to <6.5%.	- HbA1c based on ADA guidelines
6.	Interdisciplinary collaboration practice	- Teamwork entails discussion of the most appropriate treatment for patients	- Shared decision-making among patients, nurses, physicians, etc.
7.	Self-management	- Individualizing therapy so that patients can effectively self-manage their disease through increasing self-efficacy	- Diabetes SM education, self-efficacy enhancing intervention program
8.	Patient priority direct care	- Assess the individual as a whole including the complex interplay of comorbid conditions, psychosocial, functional status, and individual need	- Shared decision-making assessment tool

Note: ADA: American Diabetes Association; SM: self-management.

## Data Availability

All data used during the study are available from the corresponding author by request.

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
