# Peer review of "Precision Health Care Elements, Definitions, and Strategies for Patients with Diabetes: A Literature Review"

_ijerph, 2021, doi:10.3390/ijerph18126535_

Round 1

Reviewer 1 Report

Congratulation to the authors of this study. The present study addressed a very important issue related to Precision Health Care for diabetes patients, which has not drawn too much attention in previous studies. I only have minor comments:

  1. In the inclusion criteria, the authors wrote: “were published in English”. Do you mean the included studies were published in English for full text or you also included studies that only had English abstract?
  2. I like the way the authors organized the Results part: the subheadings made the whole part logical and easy to follow. However, I noticed that the authors combined type 1 and type 2 diabetes together when describing the results (in Table 1, you separated the type 1 and 2). I suggest the authors can compare if there is any difference in PHC elements for type 1 diabetes patients and type 2 diabetes patients (or can discuss this issue in the Discussion part).
  3. I noticed that many of the included studies were conducted in developed counties: e.g. Reference No. 19 conducted t in the Netherlands; No. 23 was in Denmark; No. 24 was in Spain; No. 34 was conducted in the US. How about PHC for diabetes in developing counties? Is it difficult to find such studies in developing countries?
  4. One last thing, I am just curious about how PHC would develop after the COVID-19 pandemic? I have read about some other studies focused on diabetes care after the COVID pandemic, as the pandemic has changed health-seeking behavior. Moreover, people with diabetes appear to be at increased risk of more severe COVID-19 infection (though evidence quantifying the risk is highly uncertain). I am very interested in how PHC would respond to these new challenges, or the concept and elements had any change among diabetes patients due to the pandemic?

Author Response

Thank you very much for the feedback received on our manuscript. Please find enclosed a copy of “Revision Notes” in response to editorial and peer review comments. The “Revision Notes” comprises a table providing responses to the reviewers’ comments.

We look forward to hearing from you soon.

Yours sincerely

Author

Reviewer 2 Report

Thank you for your work and contribution to the literature of precision medicine. I agree with the authors that there seems to be a lack of literature that defines precision medicine in diabetes. Though this literature review provides insight, it is still unclear how the concepts mentioned as part of the methods were identified a priori. It seems as though they were part of the inclusion/exclusion of literature but were not defined by the authors until the literature was synthesized (Table 2). Whether or not this is a correct interpretation, there needs to be additional clarity on how these concepts were identified. For instance, did the authors use a process or comparable definitions used in other diagnoses where precision medicine has been extensively researched/evaluated?

Author Response

(The authors gave the same response as above.)
